# Effects of Gossypol Exposure on Ovarian Reserve Function: Comprehensive Risk Assessment Based on TRAEC Strategy

**DOI:** 10.3390/toxics13090763

**Published:** 2025-09-09

**Authors:** Xiaoyan Sun, Jia Ying, Xuan Ma, Yunong Zhong, Ran Huo, Qingxia Meng

**Affiliations:** 1Center of Reproduction and Genetics, State Key Laboratory of Reproductive Medicine and Offspring Health, the Affiliated Suzhou Hospital of Nanjing Medical University, Suzhou Municipal Hospital, Gusu School, Nanjing Medical University, Suzhou 215000, China; sunxiaoyan200806@163.com; 2State Key Laboratory of Reproductive Medicine and Offspring Health, Nanjing Medical University, Nanjing 210000, China; yingj1997@163.com (J.Y.); 2024111807@stu.njmu.edu.cn (Y.Z.); 3The Affiliated Wuxi Center for Disease Control and Prevention of Nanjing Medical University, Wuxi Center for Disease Control and Prevention, Wuxi Medical Center, Nanjing Medical University, Wuxi 214000, China; maxuan@njmu.edu.cn

**Keywords:** gossypol, ovarian reserve, reproductive toxicity, TRAEC strategy

## Abstract

This study evaluated the reproductive toxicity and reversibility of gossypol exposure in female Institute of Cancer Research (ICR) mice using the Targeted Risk Assessment of Environmental Chemicals (TRAEC) framework. Mice treated with gossypol (20 mg/kg/day, 30 days) showed reduced body weight (35.90 ± 3.19 g vs. 30.26 ± 0.91 g, *p* < 0.001), depletion of primordial follicles (46.2 ± 4.8 vs. 27.5 ± 3.6, *p* < 0.01), and impaired oocyte maturation, with polar body extrusion decreasing from 65.9% to 22.6% at 60 μM (*p* < 0.0001). In the human granulosa-like tumor cell line (KGN), apoptosis increased to 91.1% at 20 μg/mL compared with 11.46% in controls (*p* < 0.0001). Proteomic profiling identified 151 differentially expressed proteins, enriched in steroidogenesis, immune regulation, and mitochondrial metabolism. After one-month withdrawal, partial morphological recovery was observed, but endocrine function remained impaired. The TRAEC risk score of 4.68 classified gossypol as a moderate reproductive toxicant. These findings indicate that gossypol damages ovarian reserve and oocyte competence, with only partial reversibility, highlighting the need for caution in its clinical use.

## 1. Introduction

Gossypol is a naturally occurring polyphenolic compound primarily derived from the seeds, roots, and stems of *Gossypium hirsutum* L., and is abundantly present in cottonseed oil and its byproducts, such as cottonseed meal [1,2]. Cottonseed meal, as a high-quality protein source, is widely used in livestock and poultry production in China [3,4]. As a result, humans may be indirectly exposed to gossypol through the consumption of animal products—including meat, eggs, and milk—derived from animals fed with cottonseed-based feed [3].

In addition to its agricultural relevance, gossypol has garnered attention for its broad pharmacological activities, including antiviral, antioxidant, and antitumor effects. Clinically, it has been used to manage gynecological conditions such as uterine fibroids, endometriosis, and dysfunctional uterine bleeding [2]. Furthermore, it acts as a natural inhibitor of oncogenic proteins like APE1 and members of the BCL-2 family, showing antineoplastic activity against various cancers, including breast, ovarian, and prostate malignancies [1,2]. Consequently, human exposure may also occur via its oral administration or therapeutic use. Pharmacokinetic studies have shown that gossypol is primarily eliminated through the gastrointestinal tract, with minimal urinary excretion, but it exhibits uneven tissue distribution, accumulating most in the liver, followed by bile, the kidneys, and the spleen [4,5].

Despite its therapeutic potential, gossypol poses significant toxicological risks. It has been shown to suppress lymphocyte proliferation, impair macrophage function, and disrupt immune responses. Additionally, it exerts neurotoxic and hepatotoxic effects, and its acute exposure has been associated with mortality in animal models [3,5,6]. Of particular concern is its reproductive toxicity. Historical epidemiological reports from China described a decade-long absence of childbirths in certain regions during the 1930s–1940s, later attributed to the dietary substitution of soybean oil with crude cottonseed oil due to economic hardship. By the 1960s, residents in several rural areas exhibited symptoms such as fatigue and burning sensations in the extremities—termed “burning fever”—followed by reproductive dysfunctions including amenorrhea in women and impotence in men, suggesting a possible link between gossypol exposure and reproductive impairment [6].

Experimental animal studies have corroborated the reproductive toxicity of gossypol in females, demonstrating impaired folliculogenesis, reduced viable follicle counts, and increased follicular atresia, resulting in disrupted estrous cycles, embryogenesis, and pregnancy outcomes [7,8]. In vitro studies further confirmed its cytotoxicity toward secondary oocytes and bovine embryos [9,10]. However, inconsistencies in gossypol study results remain. For example, Jimenez et al. observed no significant morphological changes in preantral follicles treated with gossypol, although its high doses induced the upregulation of estrogen receptors, implying potential disruption of intra-follicular steroidogenic signaling [11]. These conflicting findings highlight the need for a systematic and multidisciplinary evaluation of gossypol’s reproductive toxicity. Additionally, it is also unclear whether ovarian function can be restored following gossypol exposure.

Age is a critical factor influencing ovarian susceptibility to toxicants [12]. While the reproductive stage is characterized by active follicular development, reproductive senescence involves follicle depletion, hormonal imbalance, and reduced fertility [13]. The age-dependent effects of gossypol on ovarian function remain poorly understood. Moreover, previous reports of irreversible azoospermia in males following gossypol exposure raise concerns about its potential for permanent reproductive damage [14]. To address these issues, this study employed two exposure models—reproductive period and reproductive senescence—to examine stage-specific effects and assess the extent and reversibility of ovarian dysfunction caused by gossypol.

To ensure a systematic and evidence-based risk evaluation, this study adopted the Targeted Risk Assessment of Environmental Chemicals (TRAEC) framework. The TRAEC framework integrates four key dimensions—reliability scores, risk intensity scores, correlation scores, and integrity scores—and stratifies health risks into three levels: low (0–4), moderate (4–8), and high (8–10) [15,16,17]. By combining epidemiological data, in vivo and in vitro experiments, and mechanistic insights, the TRAEC framework offers a structured, quantitative approach for assessing reproductive risks of environmental chemicals such as gossypol, providing a robust foundation for risk mitigation in reproductive health contexts.

The objectives of our study are to investigate the risk of gossypol exposure on ovarian function, the differences in its effects between the reproductive and perimenopausal periods, the mechanisms underlying gossypol’s impact on ovarian function, and the reversibility of ovarian function recovery following gossypol exposure. We aim to provide experimental and theoretical foundations for the prevention and control of ovarian diseases associated with gossypol exposure.

## 2. Materials and Methods

### 2.1. Risk Assessment of Gossypol-Induced Ovarian Reserve Decline Based on the TRAEC Strategy

In this study, the Targeted Risk Assessment of Environmental Chemicals (TRAEC) strategy was employed to systematically evaluate the potential risk of gossypol exposure on ovarian reserve function. This assessment was conducted in accordance with the TRAEC framework, which comprises the following sequential steps: (1) identification of the core scientific question; (2) integration of evidence derived from targeted literature searches and in-house data, encompassing epidemiological, in vivo, and in vitro studies; and (3) comprehensive evaluation of the compiled evidence across four key dimensions—study reliability (rigor of design and execution), exposure-outcome correlation (strength of association), risk intensity (alignment with the research hypothesis), and evidence integrity (completeness and consistency of evidence). These criteria were quantitatively scored to assess the overall strength of evidence and classify the level of risk associated with gossypol-induced ovarian dysfunction.

### 2.2. Literature Retrieval and Screening

A systematic literature search was conducted using PubMed, Web of Science, and the China National Knowledge Infrastructure (CNKI) databases, covering all publications available up to 1 April 2025. The search strategy utilized the following keywords: “gossypol” AND (“ovary” OR “oocyte” OR “granulosa cell” OR “luteal cell”). Inclusion criteria covered in vivo, in vitro, cohort, or case–control studies reporting quantitative reproductive outcomes related to gossypol exposure. Studies lacking original data or relevant endpoints were excluded. Eligible studies were scored using the TRAEC framework.

### 2.3. Experimental Animals

Female ICR mice aged 6 and 40 weeks (*n* = 40 per group) were purchased from the Animal Center of Nanjing Medical University. Female Institute of Cancer Research (ICR) mice were obtained from the Laboratory Animal Center and housed in a Specific Pathogen-Free (SPF) environment with controlled temperature (22 ± 2 °C) and humidity (55 ± 5%) and a 12 h light/dark cycle. After a 1-week acclimation, experiments were conducted following ethical approval (ethics approval number: 2406007).

### 2.4. Experimental Design

Eighty female ICR mice were divided by age (6 and 40 weeks) and randomly assigned to control or gossypol groups (*n* = 20/group). Gossypol acetate (Shanghai YuanYe Bio, Shanghai, China) was given via oral gavage (20 mg/kg/day) for 30 days; controls received corn oil. Fifteen mice per group were euthanized one day post-treatment for acute toxicity evaluation. Five mice per group were monitored for 30 days to assess their recovery.

Mice were anesthetized with avertin (20 μL/g; Dalian Meilun Biotechnology, Dalian, China). Blood was collected via enucleation, and both ovaries were harvested. One ovary was fixed in 4% paraformaldehyde, paraffin-embedded, sectioned (5 μm), and H&E-stained for histology. The other ovary and serum were stored at −80 °C for molecular analyses.

### 2.5. Follicle Counting

H&E-stained serial ovarian sections were examined using a Nikon Ni-E microscope. Follicles were categorized as follows: primordial follicles—oocytes surrounded by a single layer of flattened granulosa cells; primary follicles—oocytes enclosed by a single layer of cuboidal granulosa cells; secondary follicles—two or more granulosa cell layers and zona pellucida; antral follicles—presence of an antral cavity and cumulus mass; and atretic follicles—degenerated oocyte, disrupted granulosa layers, or detachment. Every third section was analyzed; only follicles with visible nucleoli were counted to prevent duplication.

### 2.6. ELISA for Hormone Quantification

Serum AMH and FSH levels were measured using ELISA kits (Jianglai Bio, Shanghai, China). Assays were performed per manufacturer instructions, including serial dilution, standard/sample addition to 96-well plates, incubation with HRP-conjugated antibodies, substrate reaction, and OD detection at 450 nm using a BioTek Synergy2 reader (BioTek Instruments, Inc., Winooski, VT, USA). Hormone concentrations were calculated from standard curves (R^2^ ≥ 0.98). All samples were assayed in duplicate.

### 2.7. Oocyte Culture and Maturation

A 50 mM gossypol stock was prepared by dissolving 0.029 g of gossypol acetate in 1 mL of DMSO. Its working concentrations (0, 30, 60, and 120 μM) were made by adding 0, 0.6, 1.2, and 2.4 μL of the stock solution to 1 mL of M16 medium (Sigma-Aldrich, St. Louis, MO, USA), pre-equilibrated at 37 °C in 5% CO_2_. Ovaries from six 6-week-old ICR mice were collected after cervical dislocation. Germinal vesicle (GV)-stage oocytes were isolated under a stereomicroscope (Olympus SZX16, Tokyo, Japan) and cultured (40–50 per group) in media with respective gossypol concentrations. GVBD and PBE rates were assessed at 4 h and 14 h, respectively.

### 2.8. Immunofluorescence of Oocytes

GV-stage oocytes were collected from five 6-week-old ICR mice and treated with 0, 30, 60, or 120 μM gossypol to assess cytoskeletal and mitochondrial alterations.

Spindle assembly and Chromosome alignment: After 9 h of culture, oocytes were fixed (4% paraformaldehyde), permeabilized (0.5% Triton X-100), and blocked. Spindles were stained with anti-α-tubulin antibody (Thermo Fisher Scientific, Waltham, MA, USA), and nuclei were stained with Hoechst 33,342 (KeyGen BioTECH, Nanjing, China). Images were captured using a Zeiss LSM700 confocal microscope.

Mitochondrial distribution: After 8 h, oocytes were incubated with MitoTracker Red CMXRos (Thermo Fisher Scientific, Waltham, MA, USA) for 1 h, then fixed and imaged to assess mitochondrial localization.

### 2.9. KGN Cell Proliferation Assay (EdU)

KGN cells (from Prof. Ran Huo, Nanjing Medical University) were cultured in DMEM (Gibco, Carlsbad, CA, USA) with 10% FBS (NEWZERUM, Christchurch, New Zealand), 100 U/mL penicillin, and 100 μg/mL streptomycin (Invitrogen, Carlsbad, CA, USA) at 37 °C and 5% CO_2_.

A 40 μg/mL gossypol stock was prepared (0.002 g gossypol in 100 μL DMSO, diluted in 50 mL medium). Its working concentrations (0, 5, 10, and 20 μg/mL) were applied to cells in 24-well plates for 24 h. Proliferation was assessed using an EdU kit (Beyotime, Shanghai, China) following standard protocols: EdU incubation (2 h), fixation, permeabilization, click reaction, Hoechst 33,342 counterstaining (KeyGen BioTECH, Nanjing, China), and imaging on a Nikon Ti2-U microscope. EdU-positive nuclei were quantified to calculate proliferation rates.

### 2.10. KGN Cell Apoptosis Assay (TUNEL)

TUNEL assays were performed under the same gossypol treatment concentrations and duration as the EdU assay. After 24-h treatment, apoptosis was detected using the TUNEL Cell Apoptosis Kit (Beyotime, China). Cells were fixed in 4% paraformaldehyde, permeabilized with PBS, incubated with TUNEL reaction mix, and imaged via fluorescence microscopy. The proportion of TUNEL-positive cells was used as an index of apoptosis.

### 2.11. Proteomic Analysis of Ovarian Tissue

Ovaries from three control and three gossypol-treated 6-week-old mice were analyzed by data-independent acquisition (DIA) mass spectrometry (Majorbio Bio-Pharm, Shanghai, China). Total proteins were extracted and trypsin-digested, and the obtained peptides were analyzed using LC-MS/MS with an Astral high-resolution mass spectrometer (400–1200 *m*/*z*, DIA mode). Data were processed via Spectronaut using project-specific or library-free workflows. Peptide FDR was controlled at <1%. Differentially expressed proteins (DEPs) were defined as proteins with a fold change of ≥1.5 or ≤0.67 with *p* < 0.05. GO and KEGG analyses were performed for functional annotation.

### 2.12. Statistical Analysis

Data were analyzed using GraphPad Prism 10.0 (GraphPad Software, San Diego, CA, USA) and presented as the mean ± SD. Student’s *t*-test was used for two-group comparisons; one-way ANOVA with Tukey’s post hoc test for multiple groups. Mann–Whitney U test was applied to non-parametric data. *p* < 0.05 was considered statistically significant.

## 3. Results

### 3.1. Identification of the Scientific Question: Does Gossypol Exposure Affect Ovarian Reserve Function?

Gossypol has recently gained attention for its potent antitumor properties, making it a promising candidate for cancer therapy. Initially developed as a male contraceptive, its clinical use was discontinued due to concerns about irreversible infertility. However, data on its reproductive toxicity in female germ cells remain scarce. Thus, this study aims to address the following critical question: does gossypol exposure impair ovarian reserve function in females?

### 3.2. Evidence Collection and Integration

#### 3.2.1. Literature Search and Screening

Following a predefined strategy, 250 articles related to gossypol exposure and ovarian function were retrieved from PubMed, Web of Science, and CNKI. After removing 63 duplicates, 187 articles were screened based on inclusion and exclusion criteria. Full-text review led to the exclusion of 10 additional articles due to insufficient data or methodological flaws. Ultimately, 36 studies were included in the TRAEC-based systematic risk assessment (Figure 1).

#### 3.2.2. Overview of Included Studies

Key characteristics of the 36 included studies are summarized in Table 1 and Table 2.

Although gossypol has been studied for gynecological conditions such as endometriosis, uterine fibroids, and abnormal bleeding [7], no epidemiological studies specifically examined its impact on ovarian reserve. Among the included studies, two studies combined in vivo and in vitro experiments, resulting in 38 independent datasets: 17 in vivo animal studies and 21 in vitro cellular studies.

#### 3.2.3. Summary of In Vivo Studies

Seventeen in vivo studies were identified, utilizing various animal models including mice, rats, cows, ewes, hamsters, bats, and ducks. Specifically, six studies used rats; four used cows; two used mice; two used ewes; and one each used hamsters, bats, and ducks (Table 1). All studies evaluated the structural and functional effects of gossypol on ovarian tissue.

#### 3.2.4. Summary of In Vitro Studies

A total of 21 in vitro studies were included, employing diverse cell and tissue types (Table 2), including granulosa cells (*n* = 6); luteal cells (*n* = 6); oocytes (*n* = 4); whole ovarian tissue (*n* = 2); and one study each on theca cells, ovarian cells, isolated follicles, and oocyte–cumulus complexes. Collectively, these studies provide a comprehensive view of gossypol’s impact on ovarian cell morphology, function, hormone synthesis, and developmental competence, forming a robust evidence base for risk assessment.

### 3.3. Gossypol Exposure Induces Decline in Ovarian Reserve Function

To evaluate the effects of gossypol on ovarian reserve at different reproductive stages, female mice aged 6 weeks (young group) and 40 weeks (aged group) were orally administered gossypol daily for 30 days. Post-exposure, 6-week-old mice exhibited visible clinical symptoms, including sparse, coarse fur and hemorrhagic signs (petechiae and ecchymosis) on the tail, which were absent in controls (Figure 2A). Both age groups showed significant body weight reduction compared with their respective controls (6-week-old: control 35.90 ± 3.19 g vs. gossypol 30.26 ± 0.91 g; 40-week-old: control 49.17 ± 1.20 g vs. gossypol 46.88 ± 1.25 g) (Figure 2B). Ovarian volume and ovary-to-body weight ratios were significantly decreased following treatment, with more pronounced changes in the young group (*p* < 0.0001) and significant reductions in the aged group (*p* < 0.001) (Figure 2C,E). Histopathological analysis using H&E staining revealed marked disruption of ovarian architecture in gossypol-treated mice (Figure 2D,F). Ovaries from control animals displayed intact histology, with follicles at all developmental stages, abundant primordial follicles, and minimal atresia. In contrast, gossypol-treated ovaries exhibited reduced total follicle counts, depletion of primordial and primary follicles, increased atresia, and partially collapsed antral follicles with oocyte degeneration. Quantitative follicle analysis confirmed significant reductions in the numbers of primordial (*p* < 0.01), primary (*p* < 0.01), and secondary follicles (*p* < 0.05) in 6-week-old mice. Antral follicle counts also declined (*p* < 0.05), while atretic follicles significantly increased (*p* < 0.01). In the 40-week-old group, similar trends were observed, with decreased numbers of primordial and primary follicles (*p* < 0.05), and a 5-fold increase in the atretic follicle ratio (*p* < 0.01) compared with controls. To further assess endocrine function, serum levels of anti-Müllerian hormone (AMH) and follicle-stimulating hormone (FSH) were measured (Figure 2G,H). In both age groups, AMH and FSH levels showed a mild upward trend after gossypol treatment; however, these changes were not statistically significant (*p* > 0.05).

### 3.4. Effects of Gossypol on Mouse Oocytes and KGN Cells

#### 3.4.1. Impairment of Oocyte Maturation

Germinal vesicle breakdown (GVBD) and first polar body extrusion (PBE) rates were assessed after 4 and 14 h of in vitro culture, respectively (Figure 3A). At 4 h, control oocytes showed a GVBD rate of 90.39 ± 3.91%, whereas gossypol treatment (30, 60, and 120 μM) caused a significant, dose-dependent reduction to 76.07 ± 2.95%, 73.46 ± 3.76%, and 48.22 ± 5.68% (*p* < 0.05). At 14 h, PBE rates declined from 65.88 ± 8.44% in controls to 33.67 ± 3.22% (30 μM, *p* < 0.001), 22.62 ± 2.28% (60 μM, *p* < 0.0001), and 2.86 ± 1.17% (120 μM, *p* < 0.001), indicating substantial inhibition of maturation. Based on these findings, 60 μM was the chosen gossypol concentration for subsequent mechanistic studies.

#### 3.4.2. Disruption of Spindle Assembly and Mitochondrial Function

After 9 h of culture, control oocytes exhibited barrel-shaped spindles with correct chromosome alignment (Figure 3B). In contrast, 60 μM of gossypol significantly increased spindle abnormalities and misaligned chromosomes (71.6 ± 10.41% vs. 13.33 ± 11.55%, *p* < 0.01). Additionally, MitoTracker staining showed a dramatic reduction in mitochondrial activity (6.4 ± 1.43% vs. 28.93 ± 5.41% in controls, *p* < 0.0001) (Figure 3C). This suggests that gossypol disrupts both spindle formation and mitochondrial function, thereby impairing cytoplasmic maturation and oocyte competence.

#### 3.4.3. Inhibition of KGN Cell Proliferation and Induction of Apoptosis

EdU and TUNEL assays demonstrated that gossypol impairs proliferation and induces apoptosis in human granulosa-like tumor (KGN) cells. EdU incorporation decreased with increasing gossypol concentrations: 29.41 ± 7.42% at 5 μg/L (*p* < 0.01), 21.02 ± 0.68% at 10 μg/L, and 16.48 ± 3.06% at 20 μg/L (both *p* < 0.001) (Figure 3D). TUNEL staining revealed a dose-dependent increase in apoptosis: 18.50 ± 14.04% (5 μg/L, *p* > 0.05), 55.12 ± 2.70% (10 μg/L, *p* < 0.001), and 91.06 ± 2.23% (20 μg/L, *p* < 0.0001) compared with the control (11.46 ± 9.75%) (Figure 3E). Together, these results indicate that gossypol exerts direct cytotoxic effects on ovarian granulosa cells by inhibiting proliferation and promoting apoptosis.

### 3.5. Integrated Risk Assessment of Gossypol Exposure on Ovarian Reserve Function Based on the TRAEC Strategy

In this study, we applied a multi-evidence-based Targeted Risk Assessment of Environmental Chemicals (TRAEC) strategy to systematically evaluate the potential reproductive toxicity of gossypol on ovarian reserve function. The assessment focused on four core dimensions: reliability scores, risk intensity scores, correlation scores, and integrity scores. Each study was independently scored by two reviewers, with discrepancies resolved via discussion involving a third reviewer. This two-tiered review process enhances objectivity and scientific validity. Reliability was rated on a 1–10 scale (Figure 4A). Correlation scores are summarized in Figure 4B. Risk intensity (0–1 scale) indicates the degree to which findings support the study hypothesis (Figure 4C). Evidence count reflects the total number of studies across all dimensions. Integrity was adjusted to reflect the exclusion of epidemiological studies by redistributing weight between in vivo and in vitro evidence. The overall TRAEC-derived risk assessment yielded a composite score of 4.68 (Figure 4D), indicating moderate risk (score range: 4–8). No protective effects were observed.

In summary, this integrative TRAEC analysis classifies gossypol exposure as a moderate reproductive risk, due to its significant evidence of ovarian reserve impairment. These findings underscore the need for cautious use of gossypol in clinical settings and warrant further targeted investigations.

### 3.6. Proteomic Analysis of Ovarian Tissue Following Gossypol Exposure

To elucidate molecular mechanisms underlying gossypol-induced ovarian dysfunction, we performed a comprehensive proteomic comparison between ovaries from 6-week-old female mice treated with gossypol and age-matched controls (Figure 5). Mass spectrometry detected 8328 proteins, with reliable quantification of 8121 proteins across all samples. Sample correlation analysis demonstrated high intra-group reproducibility (R^2^ > 0.98) and clear inter-group discrimination (R^2^ < 0.5), consistent with principal component analysis (PCA) results (Figure 5A,B). Differential expression analysis revealed 151 significantly altered proteins following gossypol treatment, with 43 upregulated and 108 downregulated proteins (fold change ≥ 2 or ≤0.5; *p* < 0.05) (Figure 5C). KEGG pathway enrichment indicated that downregulated proteins primarily participate in cell adhesion, steroid hormone biosynthesis, and antigen processing/presentation pathways (Figure 5D), suggesting that gossypol disrupts ovarian endocrine, immune, and structural functions. Conversely, upregulated proteins were enriched in pathways related to protein digestion/absorption and basal cell carcinoma (Figure 5E), possibly reflecting compensatory stress responses. Figure 5F highlights the five most significantly downregulated proteins, associated with immune regulation, cell adhesion, and metabolism. Notably, glutathione synthetase (GSS)—a key enzyme in glutathione, cysteine, and methionine metabolism—suggests oxidative stress as a contributor to ovarian toxicity. Figure 5G presents the top five upregulated proteins, including the mitochondria-localized NAD(P)H pyrophosphatase NUDT13 and fatty acid-binding protein 3 (FABP3), the latter linked to PPAR signaling, indicating disruptions in lipid metabolism and energy homeostasis.

Overall, proteomic profiling indicates that gossypol exposure induces extensive alterations in critical signaling pathways—steroidogenesis, immune regulation, mitochondrial metabolism, and cell adhesion—thereby impairing ovarian cellular function and depleting ovarian reserve capacity.

### 3.7. Assessment of Reversibility of Gossypol-Induced Ovarian Reserve Decline

To evaluate the potential reversibility of gossypol-induced ovarian damage, subsets of 6-week-old and 40-week-old female mice underwent a 30-day recovery period following cessation of gossypol treatment. Throughout recovery, body weights in treated mice remained slightly lower than their age-matched controls (6 weeks: control 46.38 ± 2.39 g vs. gossypol 42.14 ± 0.83 g; 40 weeks: control 56.57 ± 1.37 g vs. gossypol 54.91 ± 1.64 g), with no further decline, suggesting systemic stabilization (Figure 6A). After one month of withdrawal, ovarian volume and ovary-to-body weight ratios did not differ significantly between treated and control mice in either age group (*p* > 0.05), indicating partial morphological recovery (Figure 6B,E). Histological evaluation showed modest increases in primordial and primary follicles and a reduction in atretic follicles in both groups compared with post-treatment levels. However, none of these follicular metrics returned to control baselines, and differences remained statistically non-significant (*p* > 0.05), indicating incomplete structural restoration (Figure 6C,F). Endocrine assessments revealed slight post-withdrawal increases in serum AMH and FSH, but these did not reach control levels or statistical significance (Figure 6G). Consequently, current data are insufficient to conclude that gossypol causes sustained endocrine disruption, warranting further investigation.

Within a one-month recovery period, gossypol-induced ovarian damage appears only partially reversible. Although ovarian structure and follicular composition showed modest improvement, endocrine and functional recovery was not significant. These results suggest that gossypol may exert lasting effects on ovarian reserve, and further research is needed to determine the extent and reversibility of its reproductive endocrine impact.

## 4. Discussion

This study systematically evaluated the effects of gossypol exposure on ovarian reserve function and its reversibility in female mice, elucidating its potential reproductive toxicity and endocrine-disrupting effects. We quantitatively assessed the reproductive toxicity of gossypol utilizing the TRAEC strategy that integrates multiple lines of evidence from epidemiological, in vivo, and in vitro studies.

Despite numerous studies reporting the reproductive effects of gossypol, results have been inconsistent [8,9,11,18,19,20,21,22,23,24,25,26,27,28,29,30,31,32,33,34,35,36,37,38,39,40,41,42,43,44,45,46,47,48,49,50], and the overall reproductive risk has not been clearly quantified. We adopted the TRAEC strategy to address this gap, offering a comprehensive, systematic, and quantitative risk assessment. Based on 36 eligible studies, we determined a moderate reproductive risk level (risk score: 4.68) associated with gossypol exposure. This integrative approach overcomes the limitations of relying on a single type of evidence, providing a more accurate toxicity profile for environmental chemicals. Notably, no epidemiological studies in human populations were identified, despite gossypol acetate’s clinical use for treating uterine fibroids and endometriosis [51,52], and its potential as an anticancer agent [1,53,54,55,56]. These findings highlight the need for future population-based investigations to determine the potential reproductive impact of gossypol in humans.

Our findings demonstrate that gossypol exposure significantly impairs ovarian function in female mice. Macroscopically, treated mice exhibited reduced body weight and smaller ovarian volume, suggesting inhibited overall growth and ovarian development. Previous studies support this observation, with Luz et al. (2018) reporting follicular atresia rates of 65–88% in rats, mice, and goats following seven days of gossypol treatment, and another study observing up to 79.4% atretic follicles in sheep [19]. Our study also showed that the number of growing follicles at all stages decreased after gossypol exposure, while the number of atretic follicles increased. In isolated oocyte experiments, 60 μM gossypol disrupted spindle formation and mitochondrial membrane potential, leading to a 63.7% decrease in polar body extrusion and increased oxidative stress [34,36]. These studies are consistent with our research findings. Granulosa cell studies indicate that gossypol inhibits proliferation and dysregulates steroidogenic enzymes (CYP11A1, HSD3B1, CYP17A1), reducing estradiol-17β and progesterone production, while leaving LH receptor expression unchanged [35,37,39]. Additionally, estrogen receptor α expression was altered [11]. Additionally, we also found that gossypol exposure induces apoptosis in granulosa cells. Together, these results suggest that gossypol impairs ovarian function through compound mechanisms involving follicle depletion, hormone disruption, and oocyte maturation failure.

Previous studies have not reported on the effects of gossypol on ovarian function in different age groups. Our research findings indicate that gossypol exposure led to notable structural damage within the ovarian tissue and significantly depleted the follicular reserve in both young and aged mice. Notably, the ovarian damage was considerably more severe in the 6-week-old group, suggesting that ovaries during the reproductive age may be more susceptible to gossypol-induced toxicity. This highlights a critical high-risk window that warrants further investigation to better understand the underlying mechanisms, potentially leading to the development of preventive measures or targeted therapies to mitigate the adverse effects of gossypol on ovarian health. Advances in omics technologies have deepened our understanding of gossypol’s ovarian effects.

Additionally, the development of omics technologies assists in elucidating the mechanisms of ovarian damage caused by gossypol. Metabolomic and redox proteomic studies in SKOV3 granulosa cells [56] and RNA-seq in primary granulosa cells [35] have been conducted, although whole-ovary omics remains unexplored. The cellular complexity of ovarian tissue—including oocytes, granulosa, and immune, stromal, and endothelial cells [57]—warrants comprehensive proteomic analysis. Our proteomic profiling revealed that gossypol disrupted pathways related to protein metabolism and carcinogenesis (upregulated), while downregulating pathways involved in cell adhesion, steroidogenesis, and antigen presentation. Notably, immune-related proteins such as OCIA2, H-2 class II antigen α-chain, and GALNT1 were significantly reduced [58,59,60], implicating immune dysregulation in ovarian impairment—an established factor in primary ovarian insufficiency [61,62]. Additionally, proteins involved in mitochondrial function and metabolism (NUDT13, FABP3) were altered, and glutathione synthetase—a key enzyme in redox homeostasis—was downregulated [63,64,65]. These findings align with prior evidence of gossypol’s impact on germ cell metabolism and redox balance [34,36].

We synthesized the existing literature on gossypol’s effects on female fertility and identified multiple mechanisms through which it impairs ovarian function. Gossypol disrupts oocyte development by impairing both nuclear and cytoplasmic processes—including spindle assembly and mitochondrial function [34,36]. It may also disturb the balance between granulosa and theca cell proliferation and apoptosis. Additionally, gossypol interferes with endocrine regulation by altering sex hormone production and disrupting estrogen receptor expression [35,37,39]. Moreover, it impairs cellular energy metabolism, increases ROS generation and lipid peroxidation, damages cellular ultrastructure, and induces apoptosis—particularly targeting mitochondria [34,35,36]. Our findings further suggest an immunoregulatory component to its action, potentially involving disruption of the hypothalamic–pituitary–ovarian (HPO) axis, leading to follicular atresia and endocrine dysfunction.

Historically, gossypol was evaluated as a male contraceptive [66], but concerns about irreversible fertility effects limited its use [14,67]. A multicenter study of 151 men treated for up to 16 weeks revealed that, one year post-treatment, only 10 of 24 participants in the 10 mg/day group exhibited recovered sperm counts, while 8 of 43 remained azoospermic [14]. A mouse study by Wang et al. found that, although short-term gossypol exposure caused reproductive and renal toxicity, these effects reversed following withdrawal [68]. In contrast, our study—which spanned over one complete mouse follicular cycle (17–19 days) [69] during both exposure and recovery—demonstrated only partial restoration of follicle numbers and hormone levels after one month, indicating that gossypol-induced ovarian damage may be partially irreversible. This may stem from structural ovarian damage (e.g., follicle depletion and increased apoptosis) that is not readily repairable and sustained hormonal imbalances hinting at long-term endocrine disturbance. These findings underscore the importance of fertility protection in women receiving gossypol acetate for gynecological or oncological conditions.

However, our study still has the following limitations: First, as this study was conducted using a murine model, its applicability to humans may be limited. Future studies should monitor fertility outcomes in women exposed to gossypol to evaluate both short- and long-term reproductive effects. Second, although proteomic analysis revealed multiple disrupted pathways, the interactions among these pathways and their specific molecular mechanisms require further investigation. Advanced approaches such as gene-knockout or overexpression models may help elucidate the roles of key regulatory molecules. Third, the recovery phase was limited to one month; extending this period could yield a more comprehensive understanding of the reversibility of gossypol-induced ovarian impairment.

## 5. Conclusions

This study applied an innovative environmental chemical risk assessment framework (i.e., TRAEC) to evaluate the effects of gossypol exposure on ovarian reserve by integrating published evidence with novel experimental data. The results demonstrate that gossypol poses a moderate reproductive risk, as it reduces ovarian reserve in female mice, impairs oocyte maturation, and disrupts pathways involved in hormone production and antigen processing. In vitro assays further confirmed that gossypol inhibits granulosa cell proliferation and induces apoptosis. Although partial recovery was observed one month after exposure cessation, residual functional impairment and endocrine alterations persisted, indicating potentially irreversible reproductive effects. Future studies should focus on human-based investigations to clarify gossypol’s long-term impact on ovarian health and explore alternative compounds—such as dextrorotary (+) -gossypol—that may offer therapeutic benefits without compromising fertility [70].

## Figures and Tables

**Figure 1 toxics-13-00763-f001:**
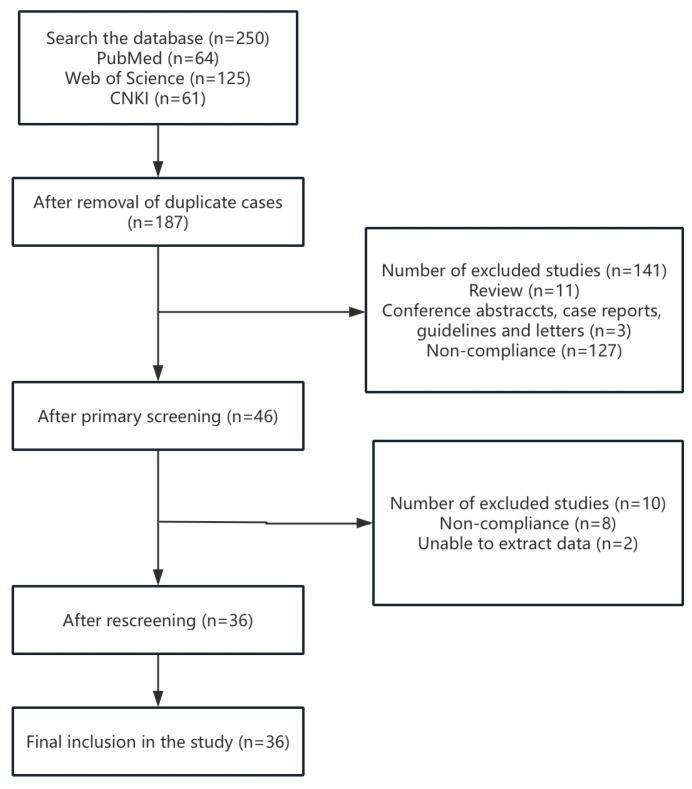
Flow diagram of selecting articles for risk assessment.

**Figure 2 toxics-13-00763-f002:**
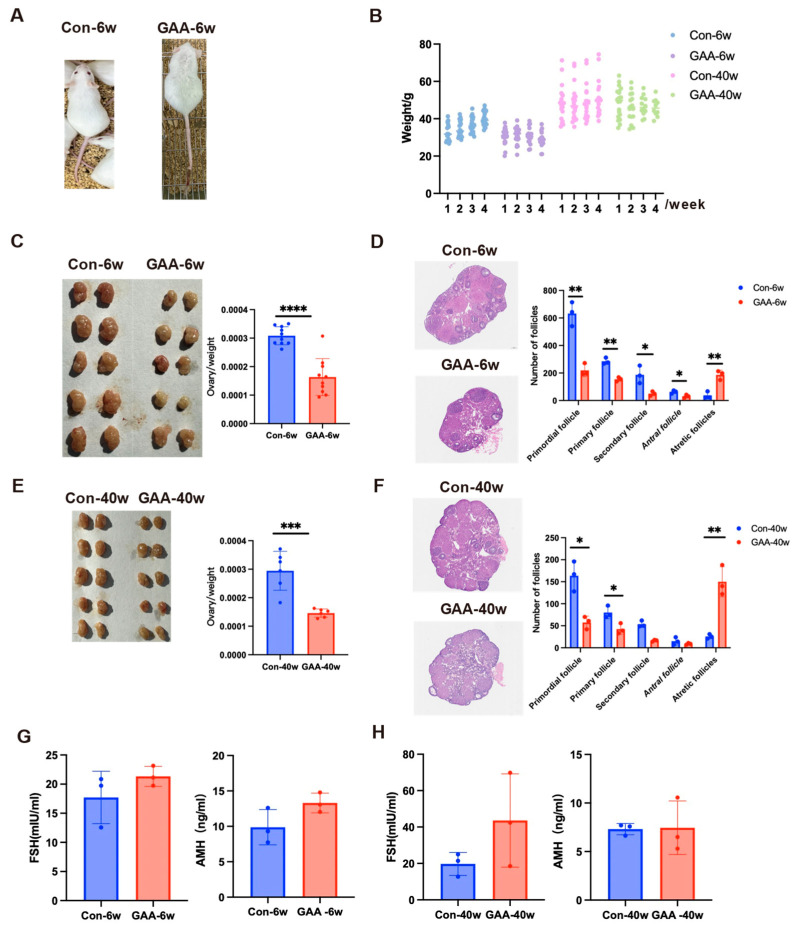
Effects of gossypol on ovarian reserve function in female mice. (**A**) Gross morphological comparison of 6-week-old mice treated with gossypol (20 mg/kg/day for 30 days) and age-matched controls, showing sparse fur and tail hemorrhagic signs in treated animals. (**B**) Body weight changes in 6-week-old control (*n* = 20) and gossypol-treated (*n* = 17) mice, and in 40-week-old control (*n* = 20) and gossypol-treated (*n* = 10) mice. (**C**) Representative images of ovarian morphology in 6-week-old control and gossypol-treated mice (*n* = 10 per group). (**D**) Hematoxylin and eosin (H&E) staining of ovarian sections from 6-week-old mice, with quantification of follicle numbers at different developmental stages (primordial, primary, secondary, antral, and atretic). (**E**) Representative ovarian morphology in 40-week-old control and gossypol-treated mice (*n* = 10 per group). (**F**) H&E staining of ovarian tissue from 40-week-old mice, with corresponding follicle counts at each stage. (**G**) Serum concentrations of follicle-stimulating hormone (FSH) and anti-Müllerian hormone (AMH) in 6-week-old control and gossypol-treated mice measured using ELISA (*n* = 3 per group). (**H**) Serum FSH and AMH concentrations in 40-week-old mice after gossypol exposure compared with controls (*n* = 3 per group). Data are presented as mean ± SD. Statistical comparisons were performed using Student’s *t*-test. **** *p* < 0.0001, *** *p* < 0.001, ** *p* < 0.01, * *p* < 0.05, and ns = not significant.

**Figure 3 toxics-13-00763-f003:**
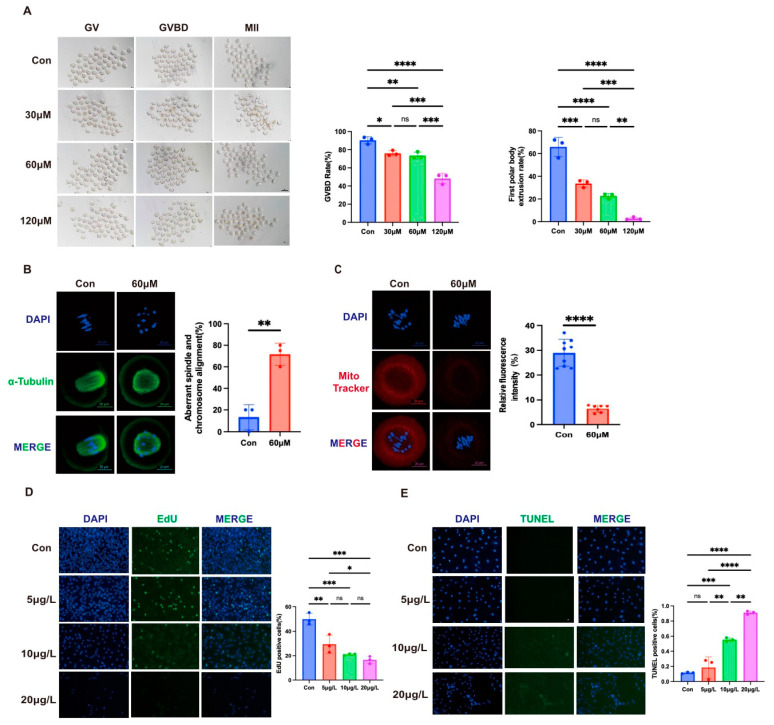
Effects of gossypol exposure on mouse oocytes and KGN cells. (**A**) Germinal vesicle breakdown (GVBD, assessed at 4 h) and first polar body extrusion (PBE, assessed at 14 h) in mouse oocytes cultured with increasing concentrations of gossypol (30, 60, and 120 μM) compared with untreated controls. Scale bar: 100 μm. (**B**) Representative confocal images of spindle morphology and chromosome alignment in oocytes cultured for 9 h with or without 60 μM gossypol. Quantification of spindle/chromosome abnormalities is shown. Scale bar: 20 μm. (**C**) Mitochondrial distribution and fluorescence intensity in oocytes cultured for 9 h with or without 60 μM gossypol, detected using MitoTracker Red staining. Scale bar: 20 μm. (**D**) Proliferation of KGN cells treated with gossypol (5, 10, and 20 μg/mL) for 24 h, measured by EdU incorporation assay. Quantitative analysis indicates dose-dependent inhibition of cell proliferation. Scale bar = 250 μm. (**E**) Apoptosis of KGN cells exposed to gossypol (5, 10, and 20 μg/mL) for 24 h, evaluated by TUNEL staining. A concentration-dependent increase in apoptosis was observed. Scale bar: 100 μm. All experiments were performed in triplicate. Data are expressed as mean ± SD. Statistical analysis was performed using one-way ANOVA. **** *p* < 0.0001, *** *p* < 0.001, ** *p* < 0.01, * *p* < 0.05, and ns—not significant.

**Figure 4 toxics-13-00763-f004:**
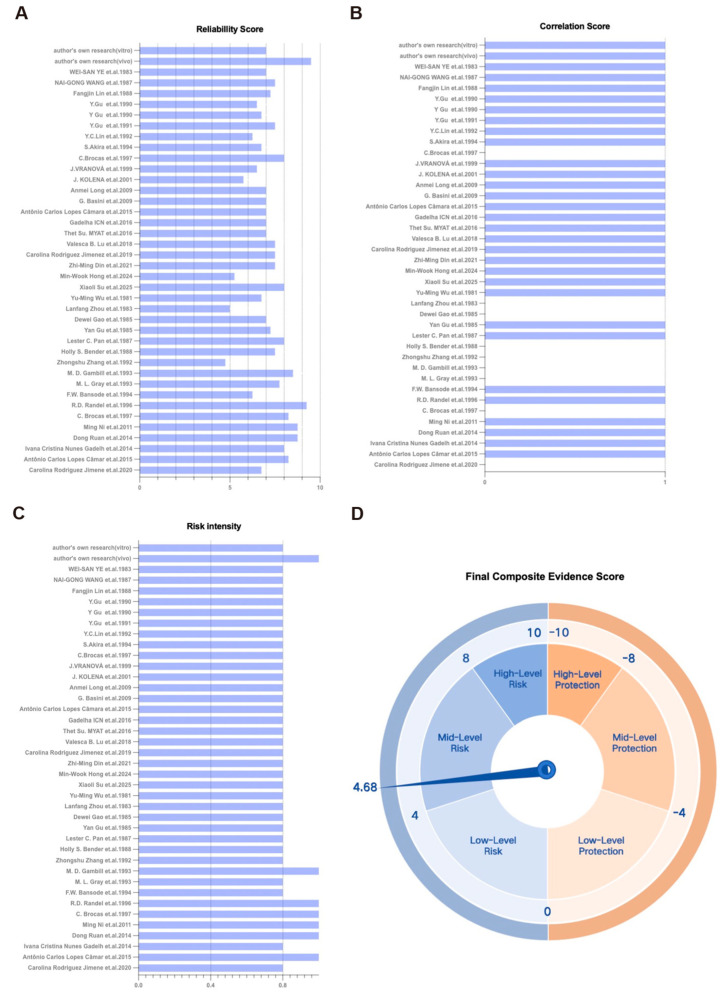
The TRAEC-based scoring results of gossypol exposure on ovarian reserve function [8,9,11,18,19,20,21,22,23,24,25,26,27,28,29,30,31,32,33,34,35,36,37,38,39,40,41,42,43,44,45,46,47,48,49,50]. (**A**) Reliability scores of the included studies, reflecting experimental design quality and methodological rigor. (**B**) Correlation scores assessing the strength of association between gossypol exposure and reproductive outcomes. (**C**) Risk intensity scores, indicating the degree to which individual studies supported the hypothesis of gossypol-induced ovarian dysfunction. (**D**) Composite evidence scores derived from integration of all four TRAEC dimensions (reliability, correlation, risk intensity, and integrity). The final analysis included both published studies (summarized in Table 1 and Table 2) and our original experimental data. Each study was independently evaluated and scored by two researchers, with discrepancies resolved through discussion with a third reviewer. Data are presented as mean values for each scoring dimension, ensuring consistency with the TRAEC framework for environmental chemical risk assessment.

**Figure 5 toxics-13-00763-f005:**
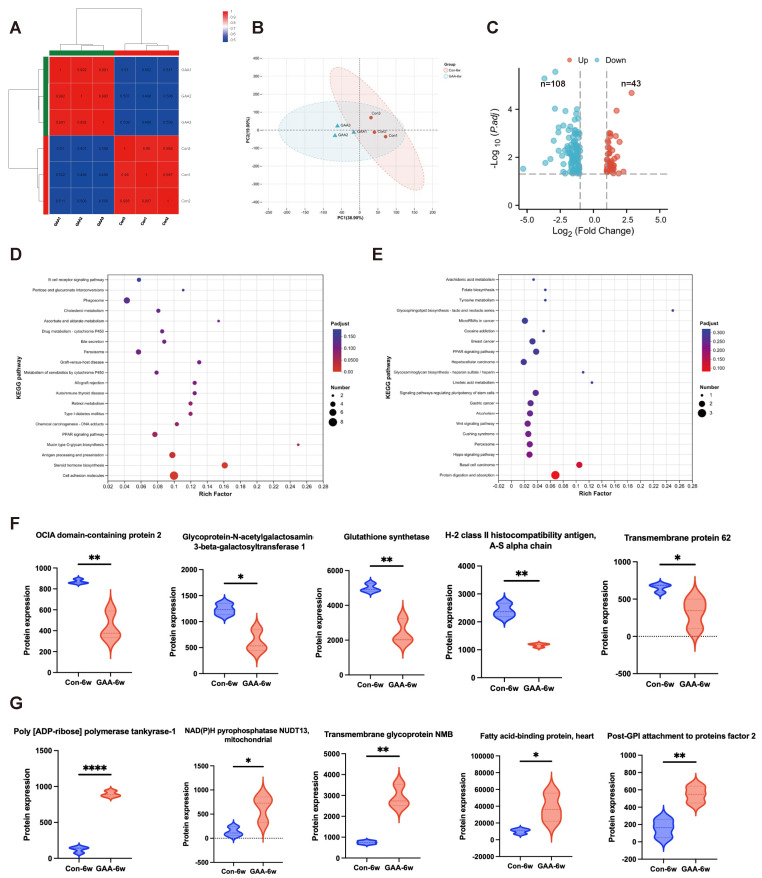
Proteomic profiling of ovarian tissue following gossypol exposure in mice. (**A**) Heatmap showing sample-to-sample correlation of quantified protein expression profiles, indicating high intra-group reproducibility and clear inter-group separation. (**B**) Principal component analysis (PCA) demonstrating distinct clustering between control and gossypol-treated ovarian tissue samples, confirming overall proteomic differences. Triangles represent the gossypol-exposed group, and circles represent the control group. (**C**) Volcano plot of differentially expressed proteins (DEPs) between gossypol-treated and control groups. DEPs were defined as proteins with fold change ≥ 2 or ≤0.5 and *p* < 0.05. (**D**) KEGG pathway enrichment analysis of significantly downregulated proteins, highlighting pathways involved in cell adhesion, steroid hormone biosynthesis, and antigen processing/presentation. (**E**) KEGG pathway enrichment analysis of significantly upregulated proteins, including pathways related to protein digestion/absorption and cellular stress responses. (**F**) The five most significantly downregulated proteins, mainly associated with immune regulation, cell adhesion, and metabolism. (**G**) The five most significantly upregulated proteins, including those involved in mitochondrial metabolism and lipid signaling. Proteomic analysis was performed using data-independent acquisition (DIA) mass spectrometry, with <1% false discovery rate for peptide identification. Data are expressed as mean values of three biological replicates per group. Statistical comparisons were conducted using *t*-tests. **** *p* < 0.0001, ** *p* < 0.01, and * *p* < 0.05.

**Figure 6 toxics-13-00763-f006:**
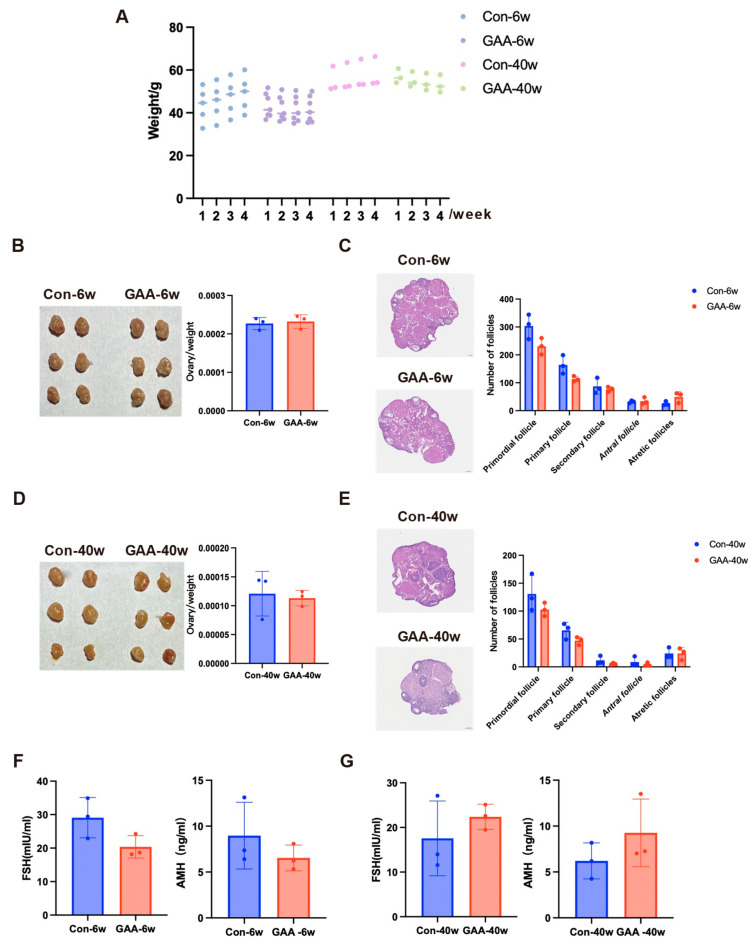
Reversibility of gossypol-induced impairment in ovarian reserve function following drug withdrawal. (**A**) Body weight changes in 6-week-old control (*n* = 5) and gossypol-treated (*n* = 7) mice, and in 40-week-old control (*n* = 3) and gossypol-treated (*n* = 3) mice, one month after cessation of gossypol administration (20 mg/kg/day for 30 days). (**B**) Representative ovarian morphology in 6-week-old control and gossypol-treated mice after one month of recovery (*n* = 6 per group). (**C**) Hematoxylin and eosin (H&E) staining of ovarian sections in 6-week-old mice after recovery, with quantitative analysis of follicle counts at different developmental stages (primordial, primary, secondary, antral, and atretic follicles). (**D**) Representative ovarian morphology in 40-week-old control and gossypol-treated mice after one month of withdrawal (*n* = 6 per group). (**E**) H&E staining and follicle count analysis in 40-week-old mice after recovery, showing partial but incomplete restoration of follicular populations. (**F**) Serum concentrations of follicle-stimulating hormone (FSH) and anti-Müllerian hormone (AMH) in 6-week-old control and gossypol-treated mice one month after drug cessation, measured by ELISA (*n* = 3 per group). (**G**) Serum FSH and AMH concentrations in 40-week-old mice after one month of withdrawal compared with age-matched controls (*n* = 3 per group). All data are presented as mean ± SD. Statistical analysis was performed using Student’s *t*-test. A partial morphological improvement was observed in both age groups, whereas endocrine parameters did not fully return to baseline. *p* < 0.05 was considered statistically significant.

**Table 1 toxics-13-00763-t001:** In vivo studies on the effects of gossypol on ovarian function in mammals.

ID	Study	Location	Species	Exposure Period	Age	Type of Gossypol	Exposure Dose	Exposure Methods	Outcomes
1	Jimenez et al., 2021 [18]	Brazil	Santa Inês ewes	10 months	/	Cottonseed	/	Feeding	Cottonseed had no effect on maternal–offspring follicular dynamics but may alter the future steroidogenic response in offspring of dams exposed to cottonseeds during their reproductive period.
2	Câmara et al., 2015 [19]	Brazil	Santa Inês crossbred ewes	63 days	/	Cottonseed cake	Feed was offered at 1.5% of the animal’s body weight.	Feeding	Gossypol-fed sheep exhibited a significantly reduced proportion of viable ovarian follicles (20.6%) and an increased proportion of atretic follicles (79.4%).
3	Gadelha et al., 2014 [20]	Brazil	Wistar rats	15 days	60 to 70 days old	Gossypol acetic acid (Fluka, G4382)	25 mg gossypol/kg/day	Subcutaneous injection	Gossypol reduced the number of viable follicles and altered hormone levels, thereby disrupting the estrous cycle.
4	Ruan et al., 2014 [21]	China	Fujian Longyan mountain hemp duck	3 months	19 weeks	Cottonseed meal	14.82, 28.03, 32.81, 39.18, and 50.16 mg/kg free gossypol	Feeding	Dominant follicle integrity was significantly compromised at a dietary free gossypol level of 28.03 mg/kg, characterized by deformation, rupture, partial dissolution, and hemorrhagic spots on follicular vascular walls.
5	Ni et al., 2011 [22]	China	Kunming mice	24 h, 48 h	25 to 30 days old	Gossypol acetic acid	1.6 and 8.40 mg/L, 0.5 mL	Intraperitoneal injection	Gossypol can effectively induce apoptosis in mouse luteal cells.
6	Brocas et al., 1997 [9]	USA	Nonlactating Holstein cow	9 weeks	/	Cottonseed meal	6.84 g free gossypol/day	Feeding	No significant differences (*p* > 0.10) were observed between cottonseed meal-fed and control cows regarding oocyte yield, cleavage rate, or blastocyst development.
7	Randel et al., 1996 [23]	USA	Brangus heifers	65 days	2 years old	Cottonseed meal; whole cottonseed	5 g free gossypol/day; 15 g free gossypol/day	Feeding	Ovarian and stromal weights, as well as total follicle counts per heifer, did not differ among the three treatment groups. However, CSM heifers had fewer follicles >5 mm compared with WCS and control heifers.
8	Bansode, 1994 [24]	India	Rhinopoma kinneari	2 days, 4 days, 6 days, 8 days	/	Gossypol acetate	10 mg/day	Feeding	Gossypol acetate exerts cytotoxic and antifolliculogenic effects, inducing oocyte and follicle degeneration and inhibiting their development.
9	Gray et al., 1993 [25]	USA	Postpubertal beef heifers; mature cows	62 days; 33 weeks	/	Cottonseed meal and whole cottonseeds	0, 0.5, 2.5, 5, 10, and 20 g free gossypol/day; 20 mg free gossypol/kg/day	Feeding	The gossypol levels used in these studies are unlikely to impair reproductive performance in beef heifers or cows.
10	Gambill and Humphrey, 1993 [26]	USA	Crossbred beef heifers	64 days	/	Cottonseed meal and whole cottonseeds	6.1 g free gossypol/day; 13.7 g free gossypol/day	Feeding	Ovarian metrics, corpus luteum characteristics, and follicle number and size were comparable across treatments.
11	Zhang et al., 1992 [27]	China	ICR mice	/	8–13 weeks	Gossypol acetic acid	5, 20, and 50 mg/kg	/	Gossypol acetate at varying doses did not affect oocyte aneuploidy rates in mice.
12	Bender et al., 1988 [28]	USA	Sprague Dawley rats	30 days	/	Gossypol acetic acid	40 and 60 mg gossypol/kg/day	Intragastric administration	Despite fewer estrous cycles, gossypol-treated rats showed no histopathological changes in their ovaries, uterus, or vagina.
13	Pan et al., 1987 [29]	USA	Rats	60 days	/	Gossypol acetic acid	20 mg gossypol/kg/day	Orally	Changes included increased ooplasmic lysosomes and underdeveloped smooth endoplasmic reticulum in granulosa cells.
14	Gu and Anderson, 1985 [30]	USA	Long–Evans strain rats	15 days; 20 days	/	Gossypol acetate	10 mg gossypol/kg/day; 1, 5, and 10 mg gossypol/kg/day	Subcutaneous injection	At male-effective doses, gossypol halted estrous cycles and significantly reduced ovarian weight.
15	Gao et al., 1985 [31]	China	Rats	Day 1–5 of gestation	/	Gossypol	100, 81, 66, 53, 43, 35, 28, and 23 mg gossypol/kg/day	Intragastric administration	Ovarian tissues showed no significant changes; corpus luteum and follicles at all stages appeared normal across all doses.
16	Zhou and Lei, 1984 [32]	China	Wistar rats	Six times a week for 8 consecutive weeks	/	Gossypol acetic acid	30 mg/kg gossypol	Intragastric administration	Ovarian morphology did not differ significantly between gossypol and control groups.
17	Wu et al., 1981 [33]	USA	Hamsters	76 days; 40 days; 20 days	/	Gossypol	5 mg gossypol/kg/day; 10 mg gossypol/kg/day; 10 and 20 mg gossypol/kg/day	Intragastric administration	Gossypol altered pituitary and ovarian hormones in proestrus and estrus but did not affect estrous length, ovulation count, or pregnancy rate.

**Table 2 toxics-13-00763-t002:** In vitro studies on the effects of gossypol on ovarian function in mammals.

ID	Study	Location	Exposure Period	Cell Type	Cell Source	Type of Gossypol	Exposure Concentration	Outcomes
1	Su et al., 2025 [34]	China	44 h	Oocytes	Porcine ovaries	Gossypol (purity >99%, MedChemExpress, Princeton, NJ, USA)	10, 20, and 40 μM gossypol	Gossypol impaired porcine oocyte maturation in vitro by reducing PB1 extrusion, inhibiting cumulus expansion, and disrupting meiosis.
2	Hong et al., 2024 [35]	Korea	72 h	Granulosa cells	Swine ovaries	/	6.25 and 12.5 μM gossypol	Gossypol is cytotoxic to porcine granulosa cells, inhibiting proliferation and impairing oocyte maturation.
3	Ding et al., 2021 [36]	China	2 h; 8 h; 14 h	Oocytes	Kunming mouse ovaries (3–4 weeks old)	Gossypol (Yirui Biotech, Hangzhou, China)	20, 40, and 60 μM gossypol	Gossypol impairs polar body extrusion, disrupts spindle structure, induces mitochondrial dysfunction and oxidative stress, and triggers early apoptosis.
4	Jimenez et al., 2019 [11]	Brazil	24 h; 96 h	Granulosa cells and oocytes	Santa Inês ewe ovaries (1-year-old)	Gossypol acetic acid (G4382, Sigma-Aldrich, São Paulo, Brazil)	5, 10, and 20 μg/mL gossypol	Gossypol impairs granulosa cell development and preantral follicle integrity in sheep.
5	Luz et al., 2018 [8]	Brazil	24 h; 7 days	Ovaries	Rat, mouse, and goat ovaries	Gossypol acetic acid (G4382, Fluka, Buchs, Switzerland)	5, 10, and 20 μg/mL gossypol	Gossypol may directly impair follicular maturation and female fertility.
6	Myat and Tetsuka, 2017 [37]	Japen	24 h at 1 day and 7 days	Theca cells	Bovine ovaries	Gossypol (Sigma-Aldrich, St. Louis, MO, USA)	0.2, 1, 5, and 25 μg/mL gossypol	Gossypol inhibits thecal steroidogenesis by downregulating steroidogenic enzyme genes without impacting cell viability in cattle.
7	Gadelha et al., 2016 [38]	Brazil	24 h; 7 days	Ovarian follicles	Adult Bantam chicken ovaries	Gossypol acetic acid (G4382, Fluka, Buchs, Switzerland)	5, 10, and 20 μg/mL gossypol	Gossypol increased atresia across all follicular stages in cultured chicken ovaries, indicating impaired follicle viability and maturation.
8	Câmara et al., 2015 [19]	Brazil	24 h; 7 days	ovaries	Santa Inês ewe ovaries (3–5 years old)	Gossypol acetic acid (G4382, Fluka, Buchs, Switzerland)	5, 10, and 20 μg/mL gossypol	Gossypol in cottonseed directly induces follicular atresia in sheep.
9	Basini et al., 2009 [39]	Italy	48 h	Granulosa cell	Large White cross-bred gilt swine ovaries	Gossypol (Sigma-Aldrich, St. Louis, MO, USA)	5 and 25 μg/mL gossypol	Gossypol markedly impairs porcine granulosa cell proliferation, steroidogenesis, and angiogenesis.
10	Long et al., 2009 [40]	China	24 h; 48 h	Luteal cells	Landrace–Yorkshire hybrid sow corpora lutea	Gossypol (Zhejiang Institute of Light Industry, Hangzhou, China)	0.4, 2, 10, and 50 mg/L gossypol	Gossypol dose-dependently inhibits luteal cell proliferation and induces apoptosis, with effects increasing over time at certain concentrations.
11	Kolena et al., 2001 [41]	Czech Republic		Oocyte–cumulus complexes	Porcine ovaries	/	10^−4^, 10^−5^, and 10^−6^ M gossypol	Gossypol suppresses FSH- and EGF-induced OCC expansion, reduces progesterone secretion, and decreases EGF receptor levels in granulosa cells.
12	Vranová et al., 1999 [42]	Slovak Republic	3 days	Granulosa cells	Porcine small follicles	Gossypol (Sigma Chemical Company, St. Louis, MO, USA)	10^−5^, 5 × 10^−5^, 10^−4^, and 5 × 10^−4^ M gossypol	Gossypol inhibits large follicles or conditioned media from stimulating progesterone production in cultured small follicles.
13	Brocas et al. (1997) [9]	USA		Oocytes	Cow ovaries	Gossypol (Sigma Chemical Company, St. Louis, MO, USA; Lot 93H4014)	2.5, 5, and 10 μg/mL gossypol	Gossypol addition during in vitro maturation did not affect oocyte cleavage rate (*p* > 0.10).
14	Akira et al., 1994 [43]	USA	48 h	Granulosa cells	Porcine ovaries	/	1–4 μM gossypol	Gossypol inhibits FSH-induced aromatase activity in cultured porcine granulosa cells.
15	Lin et al., 1992 [44]	USA	3 h	Luteal cells	Bovine	3H-gossypol acetic acid (prepared in-house following Stipanovic’s method)	2.15 and 3.4 μM 3H-gossypol acetic acid (3H-gossypol)	The cell membrane showed the highest gossypol binding, with most localized in particulate fractions.
16	Gu et al., 1991 [45]	USA	3 h	Luteal cells	Bovine ovaries containing corpora lutea	Gossypol acetic acid (Sigma-Aldrich, St. Louis, MO, USA)	4.25, 8.5, 12.75, and 17.00 μM gossypol or gossypolone	Gossypol inhibits both 3β-hydroxysteroid and cytochrome P450scc enzyme activities.
17	Gu et al., 1990a [46]	USA	3 h	Luteal cells	Dairy cow corpora lutea	Gossypol acetic acid (Sigma-Aldrich, St. Louis, MO, USA)	10, 20, and 40 μg/mL gossypol acetic acid	Gossypol dose-dependently inhibits hCG- and forskolin-induced progesterone secretion and intracellular cAMP formation.
18	Gu et al., 1990b [47]	USA	3 h	Luteal cells	Bovine ovaries containing corpora lutea	Gossypol (Sigma-Aldrich, St. Louis, MO, USA)	4.25, 8.5, 17, and 34 μM gossypol; 170 μM gossypol	Gossypol suppresses progesterone synthesis in bovine luteal cells by inhibiting steroidogenic enzymes.
19	Lin and Zheng, 1988 [48]	China	4 h; 24 h	Granulosa cells	Rats (30-day-old)	Gossypol (Shanghai Oil No. 2 Factory, Shanghai, China)	33, 44, and 55 μg/50 μL gossypol	Gossypol selectively inhibits progesterone synthesis in granulosa cells without significantly affecting aromatase activity.
20	Wang et al., 1987 [49]	China	3 h	Luteal cells	Wistar rats	Gossypol acetic acid (Sigma-Aldrich, St. Louis, MO, USA)	10, 20, and 30 μg/mL gossypol	GAA inhibits luteal steroidogenesis by suppressing gonadotropin-stimulated cAMP formation via adenylate cyclase inhibition.
21	Ye et al., 1983 [50]	USA	24 h; 48 h	Ovary cells	Chinese hamster	Gossypol acetate (Sigma-Aldrich, St. Louis, MO, USA)	5, 10, 50, and 100 μg/mL gossypol	Gossypol dose-dependently reduces survival and mitotic index of Chinese hamster ovary cells.

## Data Availability

Data is contained within the article.

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
