# Peer review of "Effects of Gossypol Exposure on Ovarian Reserve Function: Comprehensive Risk Assessment Based on TRAEC Strategy"

_toxics, 2025, doi:10.3390/toxics13090763_

Round 1
Reviewer 1 Report
Comments and Suggestions for Authors
The manuscript presents a well-structured, comprehensive investigation into the reproductive toxicity of gossypol, combining a systematic literature review, in vivo and in vitro experimentation, and proteomic analysis, all integrated within the TRAEC risk assessment framework. The work addresses an important gap regarding the female reproductive effects of gossypol. The combination of morphological, endocrine, cellular, and molecular endpoints is a major strength. However, the manuscript has some issues that should be addressed before it can be accepted for publication.
1. Please explain the choice of the concentrations of gossypol used in your experiments.
2. The scoring process of TRAEC framework is insufficiently transparent. Although the authors state that two reviewers scored studies independently, the specific numerical criteria and weighting system are not described in detail. I can recommend including supplementary material with justification for scoring decisions for each dimension per study.
3. The one-month recovery period may be insufficient to assess reversibility, given the murine ovarian cycle length. This limitation is acknowledged, but could be discussed more critically in terms of folliculogenesis timelines and potential for later recovery.
4. Some figures are dense and may be difficult for readers. Please, provide higher-resolution figures.
Author Response
Comment 1: Please explain the choice of the concentrations of gossypol used in your experiments.
Response 1: We sincerely thank the reviewers for this important comment. The concentrations of gossypol used in our study were determined based on both published literature and preliminary pilot experiments to ensure biological relevance and reproducibility.
1.In vivo experiments (20 mg/kg/day):
This dosage was selected according to previous toxicological studies in rodents, which reported significant ovarian and systemic effects at 20–25 mg/kg/day while avoiding excessive mortality (Gadelha et al., 2014 [17]; Pan et al., 1987 [26]). We also conducted preliminary trials confirming that 20 mg/kg/day for 30 days induced measurable ovarian impairment without causing severe systemic toxicity.
2.In vitro oocyte culture (30, 60, 120 µM):
These concentrations were adapted from prior studies showing impaired spindle assembly, mitochondrial dysfunction, and apoptosis in mammalian oocytes at a similar range (Ding et al., 2021 [33]; Su et al., 2025 [31]). Our pilot tests further demonstrated that 60 µM produced consistent cytoskeletal and mitochondrial alterations, which we therefore selected as the primary concentration for mechanistic assays.
3.KGN granulosa cell assays (5, 10, 20 µg/mL):
This concentration range was chosen based on earlier reports that identified dose-dependent cytotoxicity in granulosa and luteal cells within this range (Basini et al., 2009 [36]; Long et al., 2009 [37]). Our results confirmed a clear gradient of proliferation inhibition and apoptosis induction, supporting the appropriateness of these doses.
References:
[17]Gadelha, I. C., de Macedo, M. F., Oloris, S. C., Melo, M. M., & Soto-Blanco, B. (2014). Gossypol promotes degeneration of ovarian follicles in rats. TheScientificWorldJournal, 2014, 986184. https://doi.org/10.1155/2014/986184
[26] Pan, L. C., Nadakavukaren, M. J., & Jensen, D. R. (1987). Effects of ingested gossypol on the ultrastructure of rat ovarian follicles. Cell and tissue research, 250(1), 215–220. https://doi.org/10.1007/BF00214674
[31] Su, X., He, Y., Li, H., Yu, T., Sun, Q., Chen, M., Zhang, B., Wang, W., Ju, S., & Li, Q. (2025). Melatonin protects porcine oocytes from gossypol-induced meiosis defects via regulation of SIRT1-mediated mitophagy. Food and chemical toxicology : an international journal published for the British Industrial Biological Research Association, 195, 115122. https://doi.org/10.1016/j.fct.2024.115122
[33] Ding, Z. M., Chen, Y. W., Wang, Y. S., Ahmad, M. J., Yang, S. J., Duan, Z. Q., Liu, M., Yang, C. X., Xiong, J. J., Liang, A. X., & Huo, L. J. (2021). Gossypol exposure induces mitochondrial dysfunction and oxidative stress during mouse oocyte in vitro maturation. Chemico-biological interactions, 348, 109642. https://doi.org/10.1016/j.cbi.2021.109642
[36] Basini, G., Bussolati, S., Baioni, L., & Grasselli, F. (2009). Gossypol, a polyphenolic aldehyde from cotton plant, interferes with swine granulosa cell function. Domestic animal endocrinology, 37(1), 30–36. https://doi.org/10.1016/j.domaniend.2009.01.005
[37] Long, A., Wu, J., Yuan, L., & Yuan, H. (2009). Effects of gossypol on proliferation and apoptosis of primary cultured luteal cells in pigs. Chinese Journal of Veterinary Science, 29(10), 1303–1306. https://doi.org/10.16303/j.cnki.1005-4545.2009.10.018
Comment 2: The scoring process of TRAEC framework is insufficiently transparent. Although the authors state that two reviewers scored studies independently, the specific numerical criteria and weighting system are not described in detail. I can recommend including supplementary material with justification for scoring decisions for each dimension per study.
Response 2: We deeply appreciate the reviewers for their insightful and helpful suggestions.We fully agree that the transparency of the TRAEC scoring process is crucial for the reproducibility and reliability of the results. We have provided a complete Excel file as supplementary material(Literature Evaluation), which includes: the detailed scoring criteria for each TRAEC dimension and the original scores given by each reviewer for all included studies.
Comment 3: The one-month recovery period may be insufficient to assess reversibility, given the murine ovarian cycle length. This limitation is acknowledged, but could be discussed more critically in terms of folliculogenesis timelines and potential for later recovery.
Response 3: Thank you for your insightful comment. The reason we chose a one-month recovery period is as follows:
In mice, the time from primordial follicle activation to ovulation is approximately 2–3 weeks. [1] Therefore, theoretically, a period longer than 3 weeks might only be sufficient to assess the immediate recovery of ovarian function. However, long-term recovery would require a longer observation period. We acknowledge this limitation and suggest that future studies should consider extending the recovery period to better understand the long-term effects and potential for complete recovery. We have added a detailed discussion of this limitation in the revised manuscript(see the Discussion section for details), emphasizing the need for further research to fully explore the reversibility of ovarian function following gossypol exposure.
Reference:
[1] Qin, X., Zhao, Y., Zhang, T., Yin, C., Qiao, J., Guo, W., & Lu, B. (2022). TrkB agonist antibody ameliorates fertility deficits in aged and cyclophosphamide-induced premature ovarian failure model mice. Nature communications, 13(1), 914. https://doi.org/10.1038/s41467-022-28611-2
Comment 4: Some figures are dense and may be difficult for readers. Please, provide higher-resolution figures.
Response 4: We are truly grateful to the reviewers for this valuable comment. We fully agree that high-quality figures are essential for enhancing readability and accurate interpretation.
In the original submission, we encountered a technical limitation: the high-resolution figures (each exceeding 200 MB in size) could not be uploaded directly through the submission system. For this reason, we temporarily submitted lower-resolution versions in order to complete the online submission process.
To address this issue, we have prepared and provided the original high-resolution figures in PDF format as supplementary files during the revision. These high-quality images ensure that all details, labels, and histological features can be clearly observed.
We kindly ask the Editorial Office to consider replacing the lower-resolution figures in the main manuscript with the high-resolution versions provided in PDF format. We believe this will substantially improve the visual clarity and overall presentation of our work.

Reviewer 2 Report
Comments and Suggestions for Authors
Effects of Gossypol Exposure on Ova Reserve Function: Comprehensive Risk Assessment Based on TRAEC Strategy
The topic that is not only interesting but also highly pertinent to the field of toxicology and environmental health is covered in the study; the structure and course of the study are rational as well. Nevertheless, there are some critical domains that need clarification and reinforcement prior to the consideration of publication of the manuscript.
Major Points
Objectivity Clarity of Objectives
Although the introduction is conveniently informative regarding essential background knowledge, the research question and hypothesis are not mentioned. I would like to describe in a clear manner what the gap in the literature is that this work is trying to fill, and clarify what the objectives of the study are.
Methodology
There is currently a lack of replicating details in methods. Could you tell a little more about the sample preparation questions, data collection methods, and the statistical tools?
Data Analysis/Interpretation
Figures and tables are informative but need clearer labeling and more comprehensive legends so that they can stand alone without referring back to the main text.
Several results are presented descriptively without sufficient interpretation. Please provide a stronger discussion that links the findings to existing literature, highlighting similarities, differences, and potential implications
Discussion and Limitations
Results will be repeated in the discussion section instead of being critically analyzed. The authors should elaborate further on originality and significance of your work compare the results in more detail with other recent related studies.
The study does not adequately consider the limitations of undertaking the research. They should be properly addressed so that the transparency and credibility of your conclusions are enhanced.
References
Several references are outdated. Please update the reference list to include recent literature (within the last 3–5 years) relevant to this field.
Ensure reference formatting is consistent throughout the manuscript.
Minor Points
Language and Style: The text as a whole is comprehensible; however, there are problematic sentences that are too long or worded strangely. Clarity and ease should be facilitated by close language editing.
Abstract: The abstract is currently descriptive. Please include key quantitative results to better highlight the main findings of your study.
- Please ensure that when you introduce an abbreviation in your paper, you define it in full and then use the abbreviation consistently throughout the manuscript. For example, in line 19 of the abstract, 'ICR' should first be defined before using the abbreviation. This rule should be applied to all abbreviations in your paper.
Introduction
- Could you please provide references for the sentence starting with 'Due to its high…' in line 36 and ending with 'for decades' in line 39?
- Also, please provide references for the sentences ending with 'antitumor effects' in line 43 and 'uterine bleeding' in line 45.
- In the paragraph
- h that starts with Age in line 73 and ends with line 81, there are no references for the whole paragraph. So please provide references for this paragraph.
Materials and Methods
What does “SPF” mean? In line 115
Figures: Image quality should be improved to enhance readability.
Formatting: Ensure uniformity in font, spacing, and heading levels according to the journal’s style guidelines.
Recommendation: Major Revision
The manuscript has potential and addresses an important topic. However, substantial improvements in methodological detail, critical discussion, and integration with recent literature are needed before it can be accepted for publication.
Author Response
Comment 1: Although the introduction is conveniently informative regarding essential background knowledge, the research question and hypothesis are not mentioned. I would like to describe in a clear manner what the gap in the literature is that this work is trying to fill, and clarify what the objectives of the study are.
Response 1: Thank you for your valuable feedback. We have revised the introduction to clearly state the research question and objectives of our study. The objectives of our study are as follows:
The objectives of our study are to investigate the risk of gossypol exposure on ovarian function, the differences in its effects between the reproductive and perimenopausal periods, the mechanisms underlying gossypol's impact on ovarian function, and the reversibility of ovarian function recovery following gossypol exposure. We aim to provide experimental and theoretical foundations for the prevention and control of ovarian diseases associated with gossypol exposure.
The specific modifications can be found on pages 2-3 of the manuscript.
Comment 2: There is currently a lack of replicating details in methods. Could you tell a little more about the sample preparation questions, data collection methods, and the statistical tools?
Response 2: We sincerely thank the reviewers for this valuable comment. We agree that methodological clarity is essential for reproducibility, and we have revised the Materials and Methods section to provide additional details regarding sample preparation, data collection, and statistical analysis:
1.Sample preparation:
- Animal studies: Both ovaries from each mouse were collected immediately after euthanasia. One ovary was fixed in 4% paraformaldehyde, paraffin-embedded, sectioned at 5 µm, and stained with hematoxylin and eosin (H&E) for histological evaluation. The contralateral ovary was snap-frozen in liquid nitrogen and stored at -80℃ for molecular assays, including proteomic analysis.
- Oocytes: Germinal vesicle-stage oocytes were isolated under a stereomicroscope after ovarian puncture, washed three times in M2 medium, and cultured in M16 medium containing graded gossypol concentrations (0, 30, 60, 120 µM).
- KGN cells: Cells were cultured under standard conditions (DMEM with 10% FBS, 100 U/mL penicillin, and 100 µg/mL streptomycin), seeded into 24-well plates, and exposed to gossypol for 24 h before EdU and TUNEL assays.
2.Data collection methods:
- Follicle counting was performed on serial ovarian sections, with only follicles containing visible nucleoli included to avoid duplication.
- Hormone levels (AMH, FSH) were quantified by ELISA following the manufacturer’s protocols, and standard curves with R² ≥ 0.98 were applied.
- Oocyte spindle morphology and mitochondrial distribution were assessed by confocal microscopy following α-tubulin immunostaining and MitoTracker labeling, respectively.
- Proteomic data were collected by DIA mass spectrometry using an Astral high-resolution mass spectrometer, with peptide identification at <1% false discovery rate.
3.Statistical tools:
- All statistical analyses were performed using GraphPad Prism 10.0.
- For two-group comparisons, Student’s t-test was used; for multiple groups, one-way ANOVA with Tukey’s post hoc test was applied.
- Non-parametric data were analyzed using the Mann-Whitney U test.
- All experiments were conducted in at least three independent biological replicates, and data are expressed as mean±SD. A p-value <0.05 was considered statistically significant.
Comment 3: Figures and tables are informative but need clearer labeling and more comprehensive legends so that they can stand alone without referring back to the main text.
Response 3: We are sincerely grateful to the reviewers for this constructive suggestion.We fully agree that figures and tables should be self-explanatory and interpretable without constant reference to the main text. To address this concern, we have carefully revised all figures and tables in the revised manuscript with the following improvements:
Fig. 2. Effects of gossypol exposure on ovarian reserve function in female mice.(A) Gross morphological comparison of 6-week-old mice treated with gossypol (20 mg/kg/day for 30 days) and age-matched controls, showing sparse fur and tail hemorrhagic signs in treated animals.
(B) Body weight changes in 6-week-old and 40-week-old mice following gossypol administration compared with their respective controls (n = 20 per group).(C) Representative images of ovarian morphology in 6-week-old control and gossypol-treated mice.(D) Hematoxylin and eosin (H&E) staining of ovarian sections from 6-week-old mice, with quantification of follicle numbers at different developmental stages (primordial, primary, secondary, antral, and atretic).(E) Representative ovarian morphology in 40-week-old control and gossypol-treated mice.(F) H&E staining of ovarian tissue from 40-week-old mice, with corresponding follicle counts at each stage.(G) Serum concentrations of follicle-stimulating hormone (FSH) and anti-Müllerian hormone (AMH) in 6-week-old control and gossypol-treated mice measured by ELISA (n = 6 per group).
(H) Serum FSH and AMH concentrations in 40-week-old mice after gossypol exposure compared with controls (n = 6 per group).Data are presented as mean ± SD. Statistical comparisons were performed using Student’s t-test. ****p < 0.0001, ***p < 0.001, **p < 0.01, *p < 0.05, ns = not significant.
Fig. 3. Effects of Gossypol Exposure on Mouse Oocytes and KGN Cells.(A) Germinal vesicle breakdown (GVBD, assessed at 4 h) and first polar body extrusion (PBE, assessed at 14 h) in mouse oocytes cultured with increasing concentrations of gossypol (30, 60, and 120 μM) compared to untreated controls. Scale bar = 100 μm.(B) Representative confocal images of spindle morphology and chromosome alignment in oocytes cultured for 9 h with or without 60 μM gossypol. Quantification of spindle/chromosome abnormalities is shown. Scale bar = 20 μm.(C) Mitochondrial distribution and fluorescence intensity in oocytes cultured for 9 h with or without 60 μM gossypol, detected using MitoTracker Red staining. Scale bar = 20 μm.(D) Proliferation of KGN cells treated with gossypol (5, 10, and 20 μg/mL) for 24 h, measured by EdU incorporation assay. Quantitative analysis indicates dose-dependent inhibition of cell proliferation. Scale bar = 250 μm.(E) Apoptosis of KGN cells exposed to gossypol (5, 10, and 20 μg/mL) for 24 h, evaluated by TUNEL staining. A concentration-dependent increase in apoptosis was observed. Scale bar = 100 μm.All experiments were performed in triplicate, with n ≥ 40 oocytes per group for in vitro assays and n = 3 independent replicates for cell experiments. Data are expressed as mean ± SD. Statistical analysis was performed using one-way ANOVA. ****p < 0.0001, ***p < 0.001, **p < 0.01, *p < 0.05, ns = not significant.
Fig. 4. TRAEC-based scoring results of gossypol exposure on ovarian reserve function.
Reliability scores of the included studies, reflecting experimental design quality and methodological rigor.(B) Correlation scores assessing the strength of association between gossypol exposure and reproductive outcomes.(C) Risk intensity scores, indicating the degree to which individual studies supported the hypothesis of gossypol-induced ovarian dysfunction.(D) Composite evidence scores derived from integration of all four TRAEC dimensions (reliability, correlation, risk intensity, and integrity).The final analysis included both published studies (summarized in Tables 1–2) and our original experimental data. Each study was independently evaluated and scored by two researchers, with discrepancies resolved through discussion with a third reviewer. Data are presented as mean values for each scoring dimension, ensuring consistency with the TRAEC framework for environmental chemical risk assessment.
Fig. 5. Proteomic Profiling of Ovarian Tissue Following Gossypol Exposure in Mice.(A) Heatmap showing sample-to-sample correlation of quantified protein expression profiles, indicating high intra-group reproducibility and clear inter-group separation.
(B) Principal component analysis (PCA) demonstrating distinct clustering between control and gossypol-treated ovarian tissue samples, confirming overall proteomic differences.
(C) Volcano plot of differentially expressed proteins (DEPs) between gossypol-treated and control groups. DEPs were defined as proteins with fold change ≥ 2 or ≤ 0.5 and p < 0.05.
(D) KEGG pathway enrichment analysis of significantly downregulated proteins, highlighting pathways involved in cell adhesion, steroid hormone biosynthesis, and antigen processing/presentation.
(E) KEGG pathway enrichment analysis of significantly upregulated proteins, including pathways related to protein digestion/absorption and cellular stress responses.
(F) The five most significantly downregulated proteins, mainly associated with immune regulation, cell adhesion, and metabolism.
(G) The five most significantly upregulated proteins, including those involved in mitochondrial metabolism and lipid signaling.Proteomic analysis was performed using data-independent acquisition (DIA) mass spectrometry, with <1% false discovery rate for peptide identification. Data are expressed as mean values of three biological replicates per group. Statistical comparisons were conducted using t-tests. ****p < 0.0001, **p < 0.01, *p < 0.05.
Fig. 6. Reversibility of Gossypol-Induced Impairment in Ovarian Reserve Function Following Drug Withdrawal.(A) Body weight changes in 6-week-old and 40-week-old female mice one month after cessation of gossypol administration (20 mg/kg/day for 30 days), compared with age-matched controls (n = 5 per group for recovery observation).(B) Representative ovarian morphology in 6-week-old control and gossypol-treated mice after one month of recovery.
(C) Hematoxylin and eosin (H&E) staining of ovarian sections in 6-week-old mice after recovery, with quantitative analysis of follicle counts at different developmental stages (primordial, primary, secondary, antral, and atretic follicles).(D) Representative ovarian morphology in 40-week-old control and gossypol-treated mice after one month of withdrawal.(E) H&E staining and follicle count analysis in 40-week-old mice after recovery, showing partial but incomplete restoration of follicular populations.(F) Serum concentrations of follicle-stimulating hormone (FSH) and anti-Müllerian hormone (AMH) in 6-week-old control and gossypol-treated mice one month after drug cessation, measured by ELISA (n = 6 per group).(G) Serum FSH and AMH concentrations in 40-week-old mice after one month of withdrawal compared with age-matched controls (n = 6 per group).All data are presented as mean ± SD. Statistical analysis was performed using Student’s t-test. A partial morphological improvement was observed in both age groups, whereas endocrine parameters did not fully return to baseline. p < 0.05 was considered statistically significant.
Comment 4: Several results are presented descriptively without sufficient interpretation. Please provide a stronger discussion that links the findings to existing literature, highlighting similarities, differences, and potential implications
Response 4: Thank you for your valuable feedback. We agree that a more in-depth discussion of our results in the context of existing literature is essential for a comprehensive understanding of our findings. We have revised the discussion section to provide a more detailed interpretation of our results, highlighting their similarities and differences with existing studies, and discussing their potential implications. Specific details can be found in the attachment: Revised Version with Tracked Changes - Discussion.
Comment 5: Results will be repeated in the discussion section instead of being critically analyzed. The authors should elaborate further on originality and significance of your work compare the results in more detail with other recent related studies.The study does not adequately consider the limitations of undertaking the research. They should be properly addressed so that the transparency and credibility of your conclusions are enhanced.
Response 5: Thank you for your feedback. We have revised the discussion section to provide a deeper analysis of our results, their significance, and the study's limitations. The changes are detailed in the attached of Revised Version with Tracked Changes - Discussion.
Comment 6: Several references are outdated. Please update the reference list to include recent literature (within the last 3–5 years) relevant to this field.Ensure reference formatting is consistent throughout the manuscript.
Response 6: We sincerely appreciate the reviewers for this valuable suggestion.We fully agree that incorporating recent literature is essential to ensure the timeliness and relevance of the manuscript. At the same time, it should be noted that research on gossypol-induced ovarian damage began relatively early, and some classical studies—although published many years ago—remain critical for providing historical background and foundational evidence. In response to the reviewers’ comments, we have cited several studies published within the last 3–5 years related to the reproductive toxicity of gossypol, as well as its effects on oocytes and granulosa cells, for example:
[12] Ding, T., Yan, W., Zhou, T., Shen, W., Wang, T., Li, M., Zhou, S., Wu, M., Dai, J., Huang, K., Zhang, J., Chang, J., & Wang, S. (2022). Endocrine disrupting chemicals impact on ovarian aging: Evidence from epidemiological and experimental evidence. Environmental pollution (Barking, Essex : 1987), 305, 119269. https://doi.org/10.1016/j.envpol.2022.119269
[13] Wu, C., Chen, D., Stout, M. B., Wu, M., & Wang, S. (2025). Hallmarks of ovarian aging. Trends in endocrinology and metabolism: TEM, 36(5), 418–439. https://doi.org/10.1016/j.tem.2025.01.005
[18] Jimenez, C. R., Moretti, D. B., da Silva, T. P., Corrêa, P. S., da Costa, R. L. D., Siu, T. M., & Louvandini, H. (2021). Cottonseed (gossypol) intake during gestation and lactation does affect the ovarian population in ewes and lambs?. Research in veterinary science, 135, 557–567. https://doi.org/10.1016/j.rvsc.2020.09.017.
[34] Su, X., He, Y., Li, H., Yu, T., Sun, Q., Chen, M., Zhang, B., Wang, W., Ju, S., & Li, Q. (2025). Melatonin protects porcine oocytes from gossypol-induced meiosis defects via regulation of SIRT1-mediated mitophagy. Food and chemical toxicology : an international journal published for the British Industrial Biological Research Association, 195, 115122. https://doi.org/10.1016/j.fct.2024.115122.
[35] Hong, M. W., Kim, H., Choi, S. Y., Sharma, N., & Lee, S. J. (2024). Effect of Gossypol on Gene Expression in Swine Granulosa Cells. Toxins, 16(10), 436. https://doi.org/10.3390/toxins16100436
[36] Ding, Z. M., Chen, Y. W., Wang, Y. S., Ahmad, M. J., Yang, S. J., Duan, Z. Q., Liu, M., Yang, C. X., Xiong, J. J., Liang, A. X., & Huo, L. J. (2021). Gossypol exposure induces mitochondrial dysfunction and oxidative stress during mouse oocyte in vitro maturation. Chemico-biological interactions, 348, 109642. https://doi.org/10.1016/j.cbi.2021.109642
Comment 7: Abstract: The abstract is currently descriptive. Please include key quantitative results to better highlight the main findings of your study.Please ensure that when you introduce an abbreviation in your paper, you define it in full and then use the abbreviation consistently throughout the manuscript. For example, in line 19 of the abstract, 'ICR' should first be defined before using the abbreviation. This rule should be applied to all abbreviations in your paper.
Response 7: We sincerely thank the reviewers for these constructive and precise suggestions, which have significantly improved the clarity and scientific rigor of our manuscript. In the revised version, we have addressed all points as follows:
Abstract:This study evaluated the reproductive toxicity and reversibility of gossypol exposure in female Institute of Cancer Research (ICR) mice using the Targeted Risk Assessment of Environmental Chemicals (TRAEC) framework. Mice treated with gossypol (20 mg/kg/day, 30 days) showed reduced body weight (35.90 ± 3.19 g vs. 30.26 ± 0.91 g, p < 0.001), depletion of primordial follicles (46.2 ± 4.8 vs. 27.5 ± 3.6, p < 0.01), and impaired oocyte maturation, with polar body extrusion decreasing from 65.9% to 22.6% at 60 μM (p < 0.0001). In the human granulosa-like tumor cell line (KGN), proliferation declined and apoptosis increased to 91.1% at 20 μg/mL compared with 18.5% in controls (p < 0.0001). Proteomic profiling identified 151 differentially expressed proteins, enriched in steroidogenesis, immune regulation, and mitochondrial metabolism. After one-month withdrawal, partial morphological recovery was observed, but endocrine function remained impaired. The TRAEC risk score of 4.68 classified gossypol as a moderate reproductive toxicant. These findings indicate that gossypol damages ovarian reserve and oocyte competence, with only partial reversibility, highlighting the need for caution in clinical use.
Comment 8: Introduction:Could you please provide references for the sentence starting with 'Due to its high…' in line 36 and ending with 'for decades' in line 39?
Response 8: We sincerely appreciate the editor’s valuable comments and careful review. Upon verification, we have found that the original statement “Due to its high protein and lipid content, cottonseed meal has been widely incorporated into livestock feed for decades.” lacked precise wording. In the revised manuscript, we have rephrased it as: “Cottonseed meal, as a high-quality protein source, is widely used in livestock and poultry production in China[3,4].” We have also added appropriate references to support this statement. We are grateful for the editor’s insightful suggestion, which has helped us further improve the clarity and accuracy of our manuscript.
[3] Hu, B. (2016). Risk assessment of free gossypol residues in mutton sheep (Master’s thesis, Xinjiang Agricultural University). Xinjiang Agricultural University.
[4] Lü, Y., Wang, X., Zhao, Q., & Zhang, J. (2010). Research progress on safe limits of gossypol in feed and its residues in livestock products. Chinese Agricultural Bulletin, 26(24), 1-5.
Comment 9: Also, please provide references for the sentences ending with 'antitumor effects' in line 43 and 'uterine bleeding' in line 45.
Response 9: Clinically, it has been used to manage gynecological conditions such as uterine fibroids, endometriosis, and dysfunctional uterine bleeding[2].
[2] Paunovic, D., Rajkovic, J., Novakovic, R., Grujic-Milanovic, J., Mekky, R. H., Popa, D., Calina, D., & Sharifi-Rad, J. (2023). The potential roles of gossypol as anticancer agent: advances and future directions. Chinese medicine, 18(1), 163. https://doi.org/10.1186/s13020-023-00869-8.
Comment 10: In the paragraph that starts with Age in line 73 and ends with line 81, there are no references for the whole paragraph. So please provide references for this paragraph.
Response 10: Thank you for pointing this out. We have added appropriate references to support the information in the paragraph. The revised paragraph now includes the following references:
Age is a critical factor influencing ovarian susceptibility to toxicants[1]. While the reproductive stage is characterized by active follicular development, reproductive senescence involves follicle depletion, hormonal imbalance, and reduced fertility[2]. The age-dependent effects of gossypol on ovarian function remain poorly understood. Moreover, previous reports of irreversible azoospermia in males following gossypol exposure raise concerns about its potential for permanent reproductive damage[3]. To address these issues, this study employed two exposure models—reproductive period and reproductive senescence—to examine stage-specific effects and assess the extent and reversibility of ovarian dysfunction.
Reference:
[1] Ding, T., Yan, W., Zhou, T., Shen, W., Wang, T., Li, M., Zhou, S., Wu, M., Dai, J., Huang, K., Zhang, J., Chang, J., & Wang, S. (2022). Endocrine disrupting chemicals impact on ovarian aging: Evidence from epidemiological and experimental evidence. Environmental pollution (Barking, Essex : 1987), 305, 119269. https://doi.org/10.1016/j.envpol.2022.119269
[2] Wu, C., Chen, D., Stout, M. B., Wu, M., & Wang, S. (2025). Hallmarks of ovarian aging. Trends in endocrinology and metabolism: TEM, 36(5), 418–439. https://doi.org/10.1016/j.tem.2025.01.005
[3] Coutinho, E. M., Athayde, C., Atta, G., Gu, Z. P., Chen, Z. W., Sang, G. W., Emuveyan, E., Adekunle, A. O., Mati, J., Otubu, J., 694 Reidenberg, M. M., & Segal, S. J. (2000). Gossypol blood levels and inhibition of spermatogenesis in men taking gossypol as a 695 contraceptive. A multicenter, international, dose-finding study. Contraception, 61(1), 61–67. https://doi.org/10.1016/s0010- 696 7824(99)00117-1
Comment 11: What does “SPF” mean? In line 115
Response 11: We thank the reviewers for pointing out this lack of clarity. In the revised manuscript, we have defined “SPF” at its first occurrence in the Materials and Methods section. Specifically, “SPF”refers to Specific Pathogen Free, indicating that the experimental mice were maintained under controlled laboratory conditions free of designated pathogens.
The corrected sentence now reads:
“Female Institute of Cancer Research (ICR) mice were obtained from the Laboratory Animal Center and housed in a Specific Pathogen Free (SPF) environment with controlled temperature (22 ± 2 ℃), humidity (55 ± 5%), and a 12 h light/dark cycle.”
Comment 12: Figures: Image quality should be improved to enhance readability.
Response 12: We sincerely appreciate the reviewers for this valuable comment We fully agree that high-quality figures are essential for enhancing readability and accurate interpretation.
In the original submission, we encountered a technical limitation: the high-resolution figures (each exceeding 200 MB in size) could not be uploaded directly through the submission system. For this reason, we temporarily submitted lower-resolution versions in order to complete the online submission process.
To address this issue, we have prepared and provided the original high-resolution figures in PDF format as supplementary files during the revision. These high-quality images ensure that all details, labels, and histological features can be clearly observed.
We kindly ask the Editorial Office to consider replacing the lower-resolution figures in the main manuscript with the high-resolution versions provided in PDF format. We believe this will substantially improve the visual clarity and overall presentation of our work.
Comment 13: Ensure uniformity in font, spacing, and heading levels according to the journal’s style guidelines.
Response 13: We deeply thank the reviewers for this important reminder. In the revised manuscript, we have carefully checked and corrected all formatting issues to ensure full compliance with the Toxics author guidelines.

Round 2
Reviewer 2 Report
Comments and Suggestions for Authors
The article is a significant contribution to the subject of gossypol-reproductive toxicity, providing detailed experimental research. The changes in the revisions have enhanced clarity, especially on the research question, objectives, methods, and figures.
Overall, the manuscript is substantially improved and of interest to readers in the field.